# Green Synthesis of Silver Nanoparticles Using *Randia aculeata* L. Cell Culture Extracts, Characterization, and Evaluation of Antibacterial and Antiproliferative Activity

**DOI:** 10.3390/nano12234184

**Published:** 2022-11-25

**Authors:** Antonio Bernabé-Antonio, Alejandro Martínez-Ceja, Antonio Romero-Estrada, Jessica Nayelli Sánchez-Carranza, María Crystal Columba-Palomares, Verónica Rodríguez-López, Juan Carlos Meza-Contreras, José Antonio Silva-Guzmán, José Manuel Gutiérrez-Hernández

**Affiliations:** 1Department of Wood, Pulp and Paper, University Center of Exact Sciences and Engineering, University of Guadalajara, Km 15.5 Guadalajara-Nogales, Col. Las Agujas, Zapopan 45100, Jalisco, Mexico; 2Faculty of Pharmacy, Autonomous University of the State of Morelos, Av. Universidad No. 1001, Col. Chamilpa, Cuernavaca 62209, Morelos, Mexico; 3Laboratory of Basic Sciences, Faculty of Odontology, Autonomous University of San Luis Potosí, Dr. Manuel Nava No. 2, Zona Universitaria, San Luis Potosí 78290, San Luis Potosí, Mexico

**Keywords:** plant cell cultures, biological synthesis, pH-dependent synthesis, time-dependent synthesis, reducing agents, bioactive nanoparticles

## Abstract

The demand for metallic nanoparticles synthesized using green methods has increased due to their various therapeutic and clinical applications, and plant biotechnology may be a potential resource facilitating sustainable methods of AgNPs synthesis. In this study, we evaluate the capacity of extracts from *Randia aculeata* cell suspension culture (CSC) in the synthesis of AgNPs at different pH values, and their activity against pathogenic bacteria and cancer cells was evaluated. Using aqueous CSC extracts, AgNPs were synthesized with 10% (*w*/*v*) of fresh biomass and AgNO_3_ (1 mM) at a ratio of 1:1 for 24 h of incubation and constant agitation. UV-vis analysis showed a high concentration of AgNPs as the pH increased, and TEM analysis showed polydisperse nanoparticles with sizes from 10 to 90 nm. Moreover, CSC extracts produce reducing agents such as phenolic compounds (162.2 ± 27.9 mg gallic acid equivalent/100 g biomass) and flavonoids (122.07 ± 8.2 mg quercetin equivalent/100 g biomass). Notably, AgNPs had strong activity against *E. coli*, *S. pyogenes*, *P. aeruginosa*, *S. aureus*, and *S. typhimurium*, mainly with AgNPs at pH 6 (MIC: 1.6 to 3.9 µg/mL). AgNPs at pH 6 and 10 had a high antiproliferative effect on cancer cells (IC_50_ < 5.7 µg/mL). Therefore, the use of cell suspension cultures may be a sustainable option for the green synthesis of AgNPs.

## 1. Introduction

Nanotechnology has become the most effective and rapidly developing field in materials science [1]. Moreover, the use of nanoparticles (NPs) in different medical areas has increased extensively in recent years [2,3]. The well-known physicochemical and optical properties of NPs have allowed them to be used as antibacterial [4], anticancer [5], and antiviral agents [6]. However, high demand has led to the physical and chemical-based synthesis of NPs, which require the use of chemical reducing agents, mainly organic and inorganic solvents, high temperature, and vacuum conditions, which are expensive and possibly hazardous to the environment [7,8]. Nowadays, there is a growing and urgent need to develop eco-friendly processes to synthesize NPs in which toxicity is not an aspect.

The development of green and efficient methods for the synthesis of metallic NPs has become of great importance, and such methods may involve using organisms, with plants being the most appropriate choice [1,9,10]. Plants contain metabolites such as reducing sugars, terpenoids, polyphenols, alkaloids, phenolic acids, and oxidizing–reducing enzymes, which act as reducing and protective agents in the reduction of metal ions in NPs [10,11]. In fact, NPs obtained through plants are more stable, the process is faster, and a great variety of sizes and shapes can be obtained compared with the use of microorganisms [9]. Furthermore, there is a better scalability of the process for the biosynthesis of NPs [12].

Among the metallic NPs, various studies have reported on the synthesis of silver nanoparticles (AgNPs) using plant extracts with pharmacological potential [9,13,14,15]. However, the use of wild plants is not ecologically appropriate due to species decay. The use of plant cell cultures as a biotechnological tool, on the other hand, is a sustainable alternative that allows the production of sufficient amounts of plant biomass for the ecological synthesis of nanoparticles because it represents a safe, ecological, and clean method [16,17]. For instance, the biomass of some species such as *Jatropha curcas* [12], *Medicago sativa* [18], *Sesuvium portulacastrum* L. [19], *Cucurbita máxima* [20], *Lycopersicon esculentum* [21], and *Catharanthus roseus* [22] has been used to synthesize AgNPs via aqueous extracts of callus culture. In this regard, *Randia acuelata* L. (Rubiaceae) is a shrubby plant used in traditional Mexican medicine for abortion, as an antivenom or analgesic, and in liver repair [23]. Only one study of *R. aculeata* has been reported, in which strong evidence of its traditional use against snakebites has been validated, wherein it protects against muscle tissue damage and decreases damage to red blood cells [24].

To date, there are few studies that report the use of plant cell cultures for the synthesis of NPs, while for *R. aculeata*, there are no studies for this purpose. In this work, we report a biotechnological strategy using a cell suspension culture of *Randia aculeata* to synthesize silver nanoparticles (AgNPs) in an eco-friendly way; moreover, we characterize the resulting AgNPs and evaluate their potential against pathogenic bacteria and cancer cell lines.

## 2. Materials and Methods

### 2.1. Plant Material and In Vitro Culture Conditions

The *R. aculeata* L. cell suspension culture was previously established from leaf by our group, and cell cultures were maintained by periodic subculture under the same incubation conditions for 6 months [25].

### 2.2. Extracts Preparation for Synthesis of AgNPs

Fresh biomass from 24-day-old cell suspension cultures was filtered, washed with deionized water, and the excess water was removed using a vacuum pump. Briefly, biomass and deionized water were mixed in a 1:10 ratio, disrupted using an Ultra-Turrax^®^ T10 Basic homogenizer (IKA^®^) at 10,000, 15,000, and 30,000 rpm and evaluated at 5, 10, and 15 min for each stirring speed. A cell sample was stained with 0.25% Evans blue and observed under a microscope to verify membrane disintegration [26]. Then, the mixture was centrifuged at 11,000 rpm for 7 min to precipitate traces of solids from the extract. The aqueous supernatant was recovered, filtered with Whatman No. 1 filter paper, and used to synthesize AgNPs [27,28].

### 2.3. AgNP Synthesis

AgNP synthesis was carried out by using equal volumes (1:1 ratio) of aqueous extract and AgNO_3_ (Sigma-Aldrich Inc., St. Louis, MO, USA) in 1.0 mM concentration as precursor agent. This concentration was chosen based on earlier tests with three different values of AgNO_3_ (0.1, 0.5 and 1.0 mM), wherein AgNO_3_ at 1.0 mM resulted in the highest particle production [22,29]. The reaction mixture was prepared in glass tubes sealed with a screw cap, mixed slightly, and incubated in a semi-horizontally position on an orbital shaker (SEV-PRENDO, AGO 6040, Puebla, Mexico) at 115 rpm and 25 ± 2 °C for 48 h. Meanwhile, the mixture was monitored at different time points (0, 3, 5, 6, 8, 24, and 48 h) to evaluate the complete process of AgNP formation. The color change in the aqueous medium toward a dark yellow tone indicates the formation of silver nanoparticles [27,30,31]. The effect of pH variation (6, 10, or 12) in the mixture was evaluated to synthesize the AgNPs [32]. These values were adjusted using HCl or NaOH (1 N) and quantified using a Conductronic PH140 pH-meter. After the completion of incubation (48 h), all AgNPs solutions with different pH values were centrifuged at 5000 rpm for 5 min. The AgNP supernatants were recovered and sonicated for 30 min at 45 kHz using an Elma transonic TI-H-15 Ultrasonic. All AgNPs solutions with different pH were centrifuged for 5 min at 5000 rpm and the supernatant discarded. AgNPs (precipitate) were washed and resuspended in deionized water by sonication for 30 min at 45 kHz with an Elma transonic TI-H-15 Ultrasonic, then verified by UV-Vis spectrophotometry [18,33]. All experiments were performed in triplicate. These AgNPs were used for subsequent studies.

### 2.4. Characterization of AgNPs

The first characterization technique involved the UV–vis spectrophotometer, as described above. Second, dynamic light scattering (DLS) was used to estimate the average particle size. The DLS system (Malvern: Zetasizer nano S-90) was configured with a refractive index of 0.135, absorption of 3.99, viscosity of 0.8872, and 25 °C temperature [21]. Third, transmission electron microscopy (TEM) (JEM-JEOL-2100; JEOL Ltd., Peabody, MA, USA) was used to determine particle size and morphology. A 10 µL aliquot of each AgNPs sample was deposited on 400 mesh Formvar/carbon support grids and dried in a desiccator for 24 h prior to microscopic analysis. TEM was operated at 200 kV [21,29]. Finally, Fourier-transform infrared spectroscopy (FTIR) was implemented to identify the presence of proteins associated with AgNPs that give them stability by determining the characteristic functional groups, with samples analyzed using a Spectrum Two^™^ FT-IR spectrometer [21].

### 2.5. Estimation of AgNP Concentration by UV–Vis Spectroscopy

To estimate the concentration of the synthesized AgNPs, a calibration curve was prepared based on commercial silver nanoparticles of 10 nm particle size (Cat. 730785; Sigma-Aldrich, Inc.). The concentrations of the AgNPs ranged from 1 to 12 µg/mL, and the solutions (3.5 mL) were deposited into quartz cuvettes (48 mm × 12.5 mm × 12.5 mm) and analyzed at 395 nm using an UNICAM^®^ UV 500 UV–vis spectrophotometer [34]. All synthesized AgNP samples were analyzed in triplicate.

### 2.6. Estimation of Total Phenol and Flavonoid Contents

#### 2.6.1. Preparation of Extracts

Cell suspension cultures with 24-day-old and intact plant leaf biomass were used for determination of total phenolic (TPH) and total flavonoid (TFV) contents. Dry biomass (100 mg) wase ground and macerated in 20 mL of methanol under constant stirring for 24 h, and the same sample was subjected to two cycles of extraction. Extracts were filtered using syringe filter nylon membrane filtration (CHROMPURE, 0.22 µm pore size, 25 mm membrane diameter) and concentrated slightly under vacuum conditions using a Büchi^®^ RE-111 rotary evaporator (Büchi Labortechnik AG, Flawil, Switzerland), and the final sample volume was adjusted to 10 mL.

#### 2.6.2. Estimation of the Total Phenolic and Flavonoid Content

TPH content was determined using the Folin–Ciocalteu (FC) (Sigma-Aldrich, Inc.) method [35]. A 100 μL aliquot of extract was mixed with 200 μL of FC reagent and 2 mL of H_2_O and incubated at 25 ± 2 °C for 3 min in darkness. After incubation, 1 mL of Na_2_CO_3_ (20% *w*/*v*) was added to the mixture, and it was incubated again for 1 h. Subsequently, the samples were analyzed in an UNICAM UV^®^ 500 UV–vis spectrophotometer at 765 nm. Quantification was performed using a calibration curve of gallic acid at concentrations of 0 to 50 mg/L. TPH was expressed in mg equivalent of gallic acid (GAE) per 100 g of dry biomass (DW) (mg GAE/100 g DW). All determinations were made in triplicate.

The TFV was determined using the aluminum chloride method [36], for which 250 µL of extract was mixed with 1.25 mL of distilled water and 75 µL of NaNO_2_ (5%, *w*/*v*) and incubated at 25 ± 2 °C for 6 min. Then, 150 µL of AlCl_3_ (10%, *w*/*v*) was added followed by incubation for 5 min. After incubation, 0.5 mL of 1 M NaOH was added, and the volume adjusted to 2.5 mL with distilled H_2_O, and the mixture was then homogenized. The samples were read against a blank at 510 nm. Quantification was performed using a calibration curve with quercetin at concentrations of 100 to 1600 μg/mL. TFV was expressed in mg equivalent of quercetin (mg QE) per 100 g of dry biomass (mg QE/100 g DW). All determinations were made in triplicate.

### 2.7. AgNP Antibacterial Activity

The antibacterial activity of all AgNP samples was evaluated on *Escherichia coli* (ATCC 8739), *Streptococcus pyogenes* (ATCC 19615), *Pseudomonas aeruginosa* (clinical case isolates), *Staphylococcus aureus* (ATCC 6538), *Salmonella typhimurium* (ATCC 14028), and *Staphylococcus aureus-MRSA* (ATCC 43300) bacteria. The microorganisms were incubated under aerobic conditions at 37 °C. The microdilution method that was used is described in the document M100S Performance Standards for Antimicrobial Susceptibility Testing from the Clinical Laboratory Standard Institute for microdilution testing for bacteria in broth [37,38]. The initial concentrations of the synthesized AgNPs were 31.6 µg/mL (pH 6), 38.9 µg/mL (pH 10), and 57.4 µg/mL (pH 12), and these AgNPs solutions were diluted (1:1) in Mueller Hinton broth (Difco, Detroit, MI, USA). Gentamicin (GEN) was used as a reference drug (Garamycin^®^; Schering-Plough Corporation, Kenilworth, NJ, USA) evaluated at concentrations from 40 to 0.15 µg/mL. Commercial AgNPs of 10 nm particle size (Cat. 730785, Sigma-Aldrich) were used as reference nanoparticles and were evaluated from 20 to 1.25 µg/mL. The inoculum of each of the bacteria was prepared in 0.85% saline solution with adjustment according to the 0.5 tube of the McFarland Nephelometer to guarantee a count of 1.5 × 10^8^ CFU/mL, with further dilution to a final concentration of 5 × 10^5^ CFU/mL. A row of wells was used as growth control (100 µL of broth plus 100 µL of inoculum), and another row was used as a sterility control (200 µL broth). To test their potential as antimicrobial agents, each of the concentrations was supplemented with 100 µL of the bacterial inoculum and 100 µL of the sample stock solution. All assays were performed in triplicate. The plates were covered and incubated for 24 h at 37 °C. The absorbance readings obtained using Glomax Multidetection system (Promega, Madison, WI, USA) at 600 nm were analyzed. The minimum concentration of antibacterial agent capable of inhibiting bacterial growth was defined as the minimum inhibitory concentration (MIC).

### 2.8. Antiproliferative Activity of AgNPs

The AgNPs synthesized using *R. aculeata* cell culture extracts at different pH values (6, 10, and 12) and reference AgNPs (Sigma-Aldrich) were subjected to antiproliferative assays against Hep3B and HepG2 (hepatocellular), A549 (lung), and HeLa (cervical) human cancer cell lines, which were obtained from ATCC (American Type Culture Collection, Manassas, VA, USA). We also included an immortalized human hepatocyte cell line (IHH) as a control of noncancerous cells [39].

Hep3B, HepG2, and IHH cells were grown in minimum essential medium (Invitrogen, Thermo Fisher Scientific, Inc., Waltham, MA, USA), A549 cells in DMEM/F12 medium (Invitrogen, Thermo Fisher Scientific, Inc., Waltham, MA, USA) and HeLa cells in Dulbecco’s modified high-glucose Eagle’s medium (DMEM HG, Caisson Labs, Smithfield, UT, USA) supplemented with 10% (*v*/*v*) fetal bovine serum (Biowest LLC, Riverside, MO, USA) and 2 mM glutamine. All the cultures were incubated at 37 °C in a humidified atmosphere with 5% CO_2_. 

Cells, initially 8000 cells per well in a 96-well plate, were cultured for cytotoxic evaluation. The synthesized AgNPs were evaluated at concentrations from 5 to 0.3 µg/mL for AgNPs (pH 6), 6 to 0.38 µg/mL for AgNPs (pH 10), and 9.2 to 0.57 µg/mL for AgNPs (pH 12), whereas the reference AgNPs were evaluated from 8 to 0.5 µg/mL). Paclitaxel (PTX) (Sigma Aldrich; St. Louis, MO, USA) was used as positive control. Cell cultures were exposed to treatments for 48 h. For determining the number of viable cells in proliferation, we used a kit for [3-(4,5-dimethylthiazol-2-yl)-5-(3-carboxymethoxyphenyl)-2-(4-sulfophenyl)-2H-tetrazolium] inner salt MTS assay (Promega, Madison, WI, USA), following the manufacturer’s instructions. Cell viability was determined by absorbance at 450 nm using an automatic microplate reader (Promega, Madison, WI, USA. The experiments were conducted in triplicate with three independent experiments. Data were analyzed in Prism version 8.01 statistical program, and the IC_50_ values were determined by regression analysis.

## 3. Results and Discussion

### 3.1. Extraction Condition and AgNP Synthesis

Aqueous extracts of *R. aculeata* cell suspension culture (CSC) were readily obtained by cell disruption using an Ultra-Turrax^®^ at 30,000 rpm for 10 min. Colorless and translucent extracts were obtained after centrifugation at 11,000 rpm for 7 min, which were suitable for the evaluation of AgNP synthesis. After 1 h incubation, CSC extracts without AgNO_3_ did not show any change in color, but the reaction mixtures (1:1) of extracts and silver precursor visually turned from a translucent to a slightly yellow color, and as time passed, the mixture changed to a dark yellow color, which indicates the formation of silver nanoparticles [27,31,38]. The synthesis of AgNPs is due to the reduction of silver ions by reducing agents present in the extract, such as phenols and flavonoids, etc., which can transfer electrons in the silver ion reduction process (Figure 1). As described above (see Section 2.3), three different AgNO_3_ solutions (0.1, 0.5, and 1.0 mM) were used, and all the samples were subsequently analyzed in a UV–vis spectrophotometer.

The change to yellow-brown is characteristic and indicates the formation of AgNPs; this is due to surface plasmon resonance (SPR), which is a size-dependent property of NPs [30,40,41,42]. The formation of AgNPs was increased while increasing the concentration of AgNO_3_, i.e., the maximum SPR AgNPs was observed using 1 mM AgNO_3_ and a greater intensity and definition of absorbance spectra of SPR band was shown when compared with 0.1 or 0.5 mM AgNO_3_, as can be seen in Figure 2a.

In other studies, AgNPs were synthesized using aqueous extracts from *Cissus quadrangularis* stems [43] or *Azadirachta indica* leaves [29], which were analyzed between 400 and 450 nm. It has also been reported that, when using callus extract of *Catharanthus roseus* seed [22] or stem extract of *Coleus aromaticus* [14], the concentration of 1 mM AgNO_3_ was most suitable for synthesizing AgNPs in terms of yield. The biosynthesis of AgNPs with extracts of *Caesalpinía ferrea* seeds has also been reported, with SPR observation at 423 nm [44]. The same concentrations of 1.0 mM of AgNO_3_ using extracts of *Catharanthus roseus* fresh leaf [28] or *Taxus yunnanensis* callus [45] have been reported for AgNPs. In contrast, using extracts of *Medicago sativa* callus cultures, up to 0.1 M AgNO_3_ has been used to synthesize AgNPs [18]. The variation in optimal AgNO_3_ concentration may be largely due to the chemical composition of the plant, especially variations in the presence of reducing agents such as polyphenols, which often play a key role in the reduction of metal ions [46]. These same reducing agents are also directly implicated in determining the time of particle formation. The maximum absorbance of biosynthesized AgNPs after 48 h reaction time can be seen in Figure 2b. Although evident formation of AgNPs can be observed at 3 h of incubation, the reduction was followed by an accelerated formation of nanoparticles after 6 h and up to 48 h, where the reduction of silver ions appeared to terminate. Therefore, the reduction of AgNO_3_ was assumed to be complete at 48 h, as there was no further change in the absorbance of the nanoparticles as observed by UV–vis spectroscopy (see Figure 2b) [30]

The synthesis of AgNPs using aqueous extracts of saffron waste (*Crocus sativus* L.) has also been reported, with observation of an increase in intensity at 5 h of incubation without changes in the maximum wavelength [40]. With extracts of dried *Torreya nucifera* leaves, the synthesis of AgNPs began after 3 h and was completed at 24 h [47]. In a similar study of AgNP synthesis using extracts of *Sesuvium portulacastrum* L., the extract changed color to intense brown after 24 h of incubation, and then there were no significant changes; the authors also found a greater intensity in color with the callus extract than the leaf extract [19]. In another study, using *Caesalpinia ferrea* seed extracts, the synthesis of AgNPs was observed after 24 h of incubation with a maximum absorption centered at 423 nm [44]. The contact time is one of the parameters that controls the nanoparticle size, and as it increases, the surface plasmon resonance band improves because a large amount of Ag^+^ has been converted to Ag^0^; moreover, larger increases in time lead to greater decreases in absorption intensity and wavelength, indicating aggregation and a smaller size of the nanoparticles [48].

Some of the main physical and chemical parameters that affect the synthesis of AgNPs are temperature, metal ion concentration, extract content, pH, and time agitation of reaction mixture [49], with the pH value being one of the most important factors [50]. Furthermore, the NP shape and size depends on the acidity or alkalinity of the reaction mixture [42]. In this study, after optimizing the temperature, metal ion concentration, extract content, and time agitation of reaction mixture parameters, three pH values (6, 10 and 12) of the reaction mixture were evaluated. A 1:1 concentration ratio of the extract and AgNO_3_ (final concentration at 1 mM) was used with incubation for 24 h at 25 ± 2 °C and constant agitation at 115 rpm. The UV–vis analysis showed that at the three pH (6, 10, and 12) conditions, the biomass extracts of cell suspension cultures of *R. aculeata* were able to form AgNPs (Figure 3a). The synthesis of AgNPs was dependent on the pH of the reaction mixture, and as the pH increased (pH 12), the resonance band of the surface plasmon also increased, as can be seen in Figure 3b. This behavior of the surface plasmon resonance caused by the different pH is usually related to the size and size distribution of the AgNPs [51]. The closer the maximum point of the absorbance curve is to 400 nm, the smaller the average nanoparticle size, the narrower the curve, and the smaller the size dispersion. Adjusting the pH values, a plasmon resonance centered at 413 nm was obtained for pH 12, 420 nm at pH 10, and 422 nm at pH 6; this is assuming that there is a smaller size and dispersion of nanoparticles in the extracts at pH 12 than at pH 6 and 10.

In the green synthesis of AgNPs using *Tanacetum vulgare* fruit extracts, well-defined surface plasmon spectra were obtained between 452 and 546 nm with 1.0 mM AgNO_3_ at pH 6, 8, and 10 [52]. In a study conducted by [32], it was found that the *Acalypha indica* leaf extracts formed AgNPs at pH 7 within 30 min of incubation, while there was no particle formation at pH 2 and 5, and band displacement at 500 nm corresponding to the formation of agglomerations was observed at pH 9 to 13. In another study using *Cissus quadrangularis* stem extract, the synthesis of nanoparticles occurred at pH 8 and AgNO_3_ (1 mM) at 70 °C for 24 h, although the color change started at 10 min [41]. In a comparative study of the pH of the substrate (2 to 11), callus extracts of *Medicago sativa* were capable of synthesizing AgNPs at pH 10 [18].

Studies carried out with *Pistia stratiotes* extract as a reducing agent of AgNO_3_ demonstrated that under acidic conditions (pH 4, 5, and 6), surface plasmon resonance peaks appear around 330 nm, whereas in basic media (pH 9 and 10), these SPR peaks are observed at approximately 414 nm and, furthermore, the intensity of surface plasmon resonance peaks tend to increase at pH 10 [53]. Regarding the synthesis of AgNPs with oak fruit peel extract, it was shown that the synthesis rate of AgNP increases at pH 9 and subsequently decreases with higher pH [54]. Similar results were found using *Cavendish banana* peel extract, where a basic medium (pH 8) seemed to be more suitable for the formation of AgNPs, since the absorbance values increase as the pH increases [55]. Using a chemical oxidation–reduction method with silver nitrate and sodium borohydride, the pH of the solution modifies the surface charge and oxidative dissolution of AgNPs, in which there is a decrease in particle formation with increasing pH, while at high pH, there is less aggregation and higher particle stability [50].

### 3.2. Characterization of AgNPs

#### 3.2.1. Average Size of AgNPs by DLS

DLS is an optical method used for particle size determination and indicates the average size of particles suspended in a liquid medium. According to the results, the average size of the AgNPs synthesized with cell culture extracts of *R. aculeata* was influenced by pH value. Using the extract, an average estimated size of 183 nm was achieved with pH 6, 49 nm for pH 10, and 29 nm for pH 12 (Figure 4a). In the literature, AgNPs with average sizes of 37 to 52 nm in diameter are reported [12,18]. It is also reported that the particle size is directly related to the pH of extracts, i.e., with smaller particle size at alkaline pH (>9) [21]. These sizes are only estimates, since in a study carried out by [56] on *Persea americana* bark extracts, it was reported that particles measured using DLS in solution generally tend to appear larger because this method provides a hydrodynamic size of particles and any surface-bound molecules, such as metabolites and solvent molecules (mixture of nanoparticles and stabilizers) [48,49,50,51].

In addition, the chemical composition of plant extracts is a key factor for the synthesis of nanoparticles since the presence of reducing agents such as phenols and flavonoids play a key role, because they can transfer electrons in the process of reducing silver ions (Figure 1). However, a change in the chemical nature of phenols and flavonoids affects their performance in reducing metal ions. In this regard, the results obtained in this work and those reported by other authors show that the formation of silver nanoparticles is favorable in basic medium. This could be due to the ionization of functional groups of the organic molecule at high pH; that is, the acid hydroxyls of phenols and flavonoids can be deprotonated at alkaline pH, and this step is very important because an organic molecule enriched in electrons would favor the process of deprotonation of positive silver ions. Subsequently, a chelation process could take place, which would involve the carbonyl groups obtained in the oxidation–reduction reaction between the organic molecule and the positive silver ions, including other hydroxyls that have a negative charge; in this way, the formation of elemental silver nanoparticles and their stabilization would be favored. This could favor an increase in the reaction rate, and the organic molecules loosely bound to elemental silver act as a charged layer that serves as an electrostatic barrier to aggregation. Therefore, with increasing pH, the reaction rate increases and with silver chelation, aggregation is prevented, leading to a smaller nanoparticle size [48,49,50,51].

#### 3.2.2. Characterization of AgNPs by FTIR

Characterization by FTIR is a technique used to analyze metallic NPs obtained by biological synthesis, since the organic compounds associated with the NPs generate signals, mainly protein complexes known as nanoparticle coronas. The corona provides dimensional stability and allows the nanometric structure to be kept stable in the aqueous medium without the need to use surfactants to avoid agglomeration of NPs. Figure 4b shows the FTIR spectrum of the AgNPs obtained with cell suspension cultures extracts of *R. aculeata*. Characteristic signals of functional groups present in proteins are observed; at 2900 and 3280 cm^−1^, bands appear corresponding to N–H stretching of amines (I and II). A band also appears at 1630 cm^−1^ as a result of C=O stretching of amides I, in addition to 1510 cm^−1^ vibrations of N–H bending and C-N tension of amide II, immediately followed by a band at 1380 cm^−1^ that corresponds to C–N stretching, demonstrating that protein complexes are linked to AgNPs, where all these signals are more noticeable at pH 6 [21].

In the green synthesis of AgNPs using extracts from *Plectranthus amboinicus* leaves, it is reported that in the 1620 cm^−1^ band, the absorbance at 1332 and 1226 cm^−1^ is attributed to the carboxyl group of the C-C stretching vibration present in the extract. These are associated with C-O belonging to polysaccharides and C-O belonging to polyols, respectively, which can be used for the reduction of metal ions to nanoparticles [57]. In the green synthesis of AuNPs using *Curcuma pseudomontana* extracts, bands around 1450–1580 cm^−1^ have also been found, characteristic of benzene in the essential oil; the bands at 1400 and 1700 cm^−1^ belong to carbonyl groups such as carboxylate residues and ketone [58]. This means that surfactants are not necessary for this synthesis process, which is essential when traditional physical or chemical methods are used, helping to provide stability and avoid agglomerations of the NPs in different media [14].

#### 3.2.3. Morphology and Size Analysis of AgNPs by TEM

In this study, TEM analysis showed that the size of AgNPs was smaller than that estimated by DLS. Moreover, elemental analysis by EDX confirmed the presence of silver in the sample (Appendix A). The AgNPs synthesized at pH 12 were able to observe polydisperse particles with sizes from 10 to 60 nm in diameter, mainly individual particles (Figure 5a,b).

In the AgNPs obtained at pH 10, polydisperse, spherical particles and a few agglomeration zones were also observed, but with larger sizes ranging from 40 to 90 nm (Figure 5c,d). Regarding the AgNPs synthesized at pH 6, polydisperse particles without agglomerations were observed in a size range from 10 to 40 nm (Figure 5e,f). The TEM analysis indicates that AgNPs of a smaller diameter were obtained with the pH 6 extracts, contrary to what is indicated by DLS analysis. This is because the DLS does not allow differentiating between individual metallic particles and agglomerated particles in addition to the fact that this method provides a hydrodynamic size [12,56]. These authors reported that high temperatures promote more rapid synthesis of silver nanoparticles, with mainly spherical NPs being obtained. In the reaction mixture with extracts of *R. aculeata* incubated at 25 ± 2 °C, under constant agitation for 24 h and at pH 6, spherical AgNPs with sizes ranging from 10 to 40 nm were obtained. This study thus demonstrates that high temperatures are not necessary to obtain spherical NPs of optimal size. In addition, extracts from a biotechnological and sustainable crop were used in this study.

### 3.3. Estimation of the Concentration of AgNPs by UV–Vis

The estimation of NP concentration by biological synthesis has scarcely been reported due to it being relatively complicated when compared with the direct method; in addition, there is high consumption of reagents. However, in this study, indirect estimates were conducted using UV–vis spectrophotometry and a standard solution of AgNPs. Based on the calibration curve of commercial AgNPs with concentrations ranging from 1 to 10 µg/mL, a dependence between the absorbance and concentrations is observed, i.e., as the concentration of AgNPs increases, the absorbance also increases. In this study we found an R^2^ = 0.9999 (Appendix A). In their study, Hung et al. [34] also found a direct relationship between the concentration and the absorbance values, concluding that when there are R^2^ values greater than 0.99 in the calibration curve, it can be used to reliably estimate the concentration of AgNPs. In our study, using extracts from cell cultures of *R. aculeata* and 1 mM AgNO_3_ at a 1:1 ratio of extract and reagent, respectively, the AgNPs obtained at pH 6 had a concentration of 31.64 µg/mL, 38.97 µg/mL at pH 10, and 57.4 µg/mL at pH 12. Considering these concentrations, the evaluation of antibacterial and antiproliferative activity on cancer cells was carried out (Section 3.5 and Section 3.6).

### 3.4. Total Phenolic and Flavonoid Content of the Extracts

It has been reported that plant extracts are able to function as reducing agents for the synthesis of AgNPs, since the extracts contain mainly flavonoids and phenolics [59]. In our study, statistical analysis showed that leaf extracts and suspension cell cultures of *R. aculeata* had no significant differences (*p* ≤ 0.05) in total phenol and flavonoid content (Table 1). Regarding the content per group of compounds, both types of extracts (leaf or cell suspension cultures) exhibited a greater amount of total phenolics than flavonoids. In a previous study, we also reported that the aqueous extracts of *Randia aculeata* produced important amounts of saponins [25], which could also be contributing to the synthesis of AgNPs as a stabilizing agent [60].

In studies carried out in methanolic extracts of *Randia nitida* leaves, 524.5 mg GAE/g DW of total phenolic and 178.5 mg QE/g DW of flavonoid content were reported [61]. These values are slightly higher than those for *R. aculeata* in our study. In contrast, in *Randia dumetorum*, low values of total phenolics (112 ± 3.24 mg GAE/g extract) and flavonoids (2.6 ± 0.26 mg routine equivalent/g extract) were reported [62]. On the other hand, the analysis of *Randia echinocarpa* extracts showed high amounts of total phenolics in the pulp fruit, with 464 mg GAE/g in pulp. These values are higher than that found in *R. aculeata* leaves, since they are different species [63]. In fact, the presence of compounds in the same plant species can be influenced by intrinsic and extrinsic factors such as environment, weather, topography, and extraction method [64].

### 3.5. Antibacterial Activity of AgNPs

The AgNPs synthesized with extracts of *R. aculeata* at different pH were evaluated on five pathogenic bacterial strains. All the synthesized AgNPs had an effect against all the tested bacteria (MIC ≤ 14.3 µg/mL), but AgNPs synthesized with the extracts at pH 6 (10 to 40 nm) showed the highest antibacterial activity, with an MIC ranging from 1.9 to 3.9 µg/mL. *Pseudomonas aeruginosa* was the strain that had the highest sensitivity, mainly with AgNPs obtained at pH 6 (MIC = 1.9 µg/mL), followed by that obtained at pH 10 (MIC = 2.4 µg/mL) and then pH 12 (MIC = 3.5 µg/mL). Commercial AgNPs used as controls were also able to inhibit the growth of *P. aeruginosa*, although the MIC = 5 µg/mL is higher than those of the synthesized AgNPs. Moreover, it was observed that AgNPs synthesized with *R. aculeata* were more effective against *Streptococcus pyogenes*, *P. aeruginosa*, and *Staphylococcus aureus*-MRSA than gentamicin, which was the positive control (Table 2).

It is known that several pathogenic bacteria have developed resistance against various antibiotics, but it has also been reported that silver nanoparticles have remarkable antimicrobial potential and can be usefully applied in silver-based dressings and silver-coated medical devices [65]. This antibacterial activity may be due to the release of silver ions and/or the specific functions of the particle; that is, the nanoparticles and silver ions bind to the cell wall and membrane, causing damage to biomolecules and intracellular structures, in addition to causing oxidative stress [66,67,68,69]. In a study conducted by Keshari et al. [70], *Cestrum nocturnum* extracts were used to synthesize AgNPs with an average size of 20 nm, mostly spherical in shape, which had strong antibacterial activity mainly against *Enterococcus faecalis* (MIC = 4 µg/mL), *Escherichia coli* (MIC = 8 µg/mL), and *Proteus vulgaris* (MIC = 8 µg/mL). On the other hand, based on inhibition zone assays, AgNPs (7–14 nm) synthesized using *Thuja occidentalis* leaf extracts were found to have inhibitory effect against *S. aureus* and *E. coli* at 40 µL of plant extract/nanoparticles, while leaf extracts were not effective [71]. By using AgNPs synthesized with extracts of *Plumeria alba*, it was found that they showed good antibacterial activity at concentrations of 25–100 µg/mL in terms of zone of inhibition, in which large zones of inhibition were observed mainly for *Streptococcus pneumoniae* (16 mm) at 100 µg/mL [72]. Similarly, other studies report obtaining AgNPs (approximately 15 nm in diameter) using waste material (dry grass), demonstrating activity against *P. aeruginosa* and *Acinetobacter baumannii* with a MIC = 3 µg/mL of AgNPs [73]. On the other hand, using callus culture extracts of *Cinnamonum camphora*, spherical AgNPs with an average size of 5.47 to 9.48 nm have also been synthesized, which exhibited broad-spectrum activities against *S. aureus*, *Bacillus subtilis*, *E. coli*, and *P. aeruginosa*, with zones of inhibition ranging from 15 to 19.6 mm [16]. For AgNPs obtained with extracts from *Sesuvium portulacastrum* callus cultures, inhibition zones of 15 to 18 mm are reported for *P. aeruginosa*, *Klebsiella pneumoniae*, *S. aureus*, *Listeria monocytogenes*, and *Micrococcus luteu* [19]. It has also been reported that silver nanoparticles from *Paederia foetida* L. leaf extracts exhibit strong activity against *Bacillus subtilis*, *Bacillus cereus*, *Escherichia coli*, and *Pseudomonas aeruginosa*, showing inhibitory zones of bacterial growth [74].

### 3.6. Antiproliferative Activity of AgNPs

There are several studies on the ecological synthesis of AgNP for various applications, particularly in cancer treatment or with antiproliferative activity, proving to be better candidates in terms of size, drug loading and release efficiency, targeting efficiency, minimal side effects associated with the drug, pharmacokinetic profile, and biocompatibility issues [75]. In this study, the in vitro antiproliferative activity of the AgNPs synthesized with extracts of *R. acuelata* cell cultures were evaluated at concentrations of 0.3 to 9.2 µg/mL using the human cancer cell lines Hep3b, HepG2, HeLa, and A549, with the IHH line used as a control (Table 3).

All AgNPs synthesized at different pH had a significant antiproliferative effect on all cancer lines, with IC50 ranging from 1.2 to 7.6 µg/mL. Regarding antibacterial activity, the effect of AgNPs on antiproliferative activity was dependent on the pH at which they were synthesized, i.e., there was a greater antiproliferation with AgNPs obtained with pH 6 than pH 10 or 12; this behavior was observed for all cell lines, including the IHH line. Hep3b and HepG2 cells showed the lowest values with IC_50_ = 1.2 µg/mL. In Appendix A, some morphological changes, growth inhibition, rounding, and cell detachment are observed in the Hep3B cell line, which are characteristic of the effect of the treatments [76,77]. Although the IHH cell line was also affected, its IC_50_ values were greater than 3.0 µg/mL. Compared with commercial AgNPs, these showed IC_50_ values higher than the nanoparticles obtained at pH 6 (Table 3).

In a study carried out for the synthesis of AgNPs using the aqueous extract of *Cucumis prophetarum* leaves, the antiproliferative activity in cancer cell lines A549, MDA-MB-231 (breast), HepG2, and MCF-7 (breast) was evaluated, in which the IC_50_ values were 105.8, 81.1, 94.2, and 65.6 μg/mL of AgNPs, respectively, showing that AgNPs were more selective toward MCF-7 [78]. In another study, AgNPs (17 to 40 nm in size) obtained with aqueous extracts from *Galphimia glauca* leaves demonstrated strong antiproliferative activity against the SK-HEP1 (liver) cell line, with an IC_50_ value of 19.12 µg/mL of AgNPs [79]. AgNPs were synthesized using grass waste extracts, which were evaluated against the cancerous cell line MCF-7; an inhibitory effect on growth of 30% was achieved when using 5 µg/mL of AgNPs [73]. Through a biotechnological study, AgNPs were obtained using biomass from *Ocimum basilicum* leaf callus cultures, and their antiproliferative potential against HepG2 liver carcinoma cells was demonstrated, with the finding that AgNPs synthesized with callus anthocyanins were more effective (24.7%) than those obtained using crude callus extracts at 200 µg/mL [80].

## 4. Conclusions

There are several studies on the synthesis of plant nanoparticles using plant extracts or their waste, which are ecologically nontoxic; however, the variability of each species has led to different results due to the great diversity in chemical compositions. For the first time, we report the use of colorless aqueous extracts from suspension cell culture biomass of *R. aculeata* as a rapid and sustainable biotechnological resource for the green synthesis of plant nanoparticles. Furthermore, at low concentrations, the synthesized AgNPs were able to inhibit the growth of human pathogenic bacterial strains (MIC ≤ 14.3 µg/mL) and cancer cell lines (IC50 ≤ 7.6 µg/mL). Moreover, use of a plant cell culture favors obtaining homogeneous extracts and allows better control of the synthesis of AgNPs during all seasons of the year.

## Figures and Tables

**Figure 1 nanomaterials-12-04184-f001:**
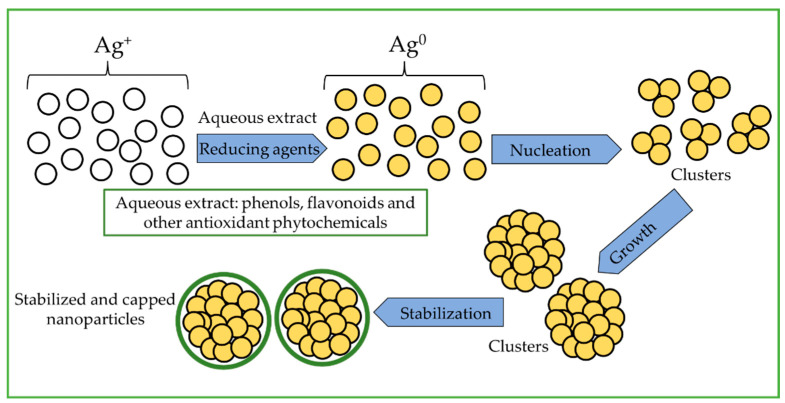
General scheme of the synthesis mechanism of silver nanoparticles using plant extracts.

**Figure 2 nanomaterials-12-04184-f002:**
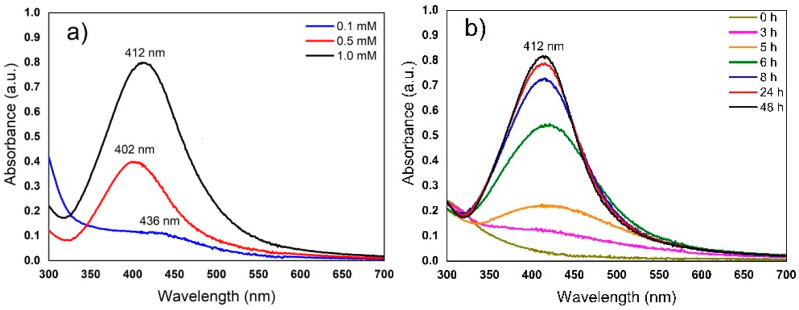
UV–vis spectra of synthesized AgNPs incubated at 25 ± 2 °C under constant shaking at 115 rpm for 24 h. (**a**) Effect of the concentration ratio of AgNO_3_ and extract (1:1); (**b**) AgNP synthesis with 1 mM AgNO_3_ and extract (1:1) at different time intervals for 48 h.

**Figure 3 nanomaterials-12-04184-f003:**
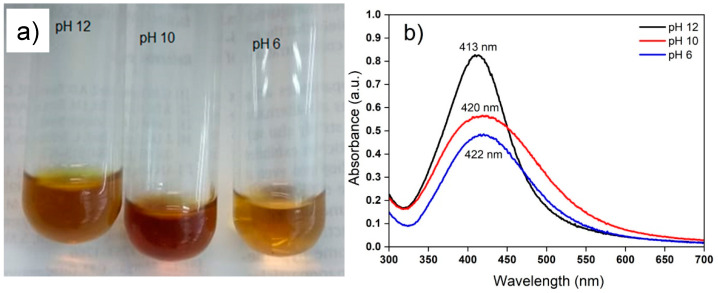
Effect of the extract pH on the formation of AgNPs monitored by UV–vis at 24 h. (**a**) Typical coloration of AgNPs synthesized and (**b**) absorption spectra at different pH.

**Figure 4 nanomaterials-12-04184-f004:**
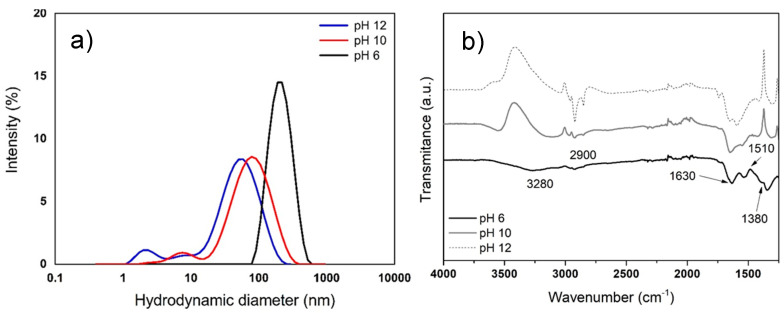
Characterization of AgNPs synthesized using aqueous extracts of *R. aculeata* cell suspension culture at different pHs. (**a**) DLS analysis of the average size of AgNPs; (**b**) FTIR spectra of synthesized AgNPs.

**Figure 5 nanomaterials-12-04184-f005:**
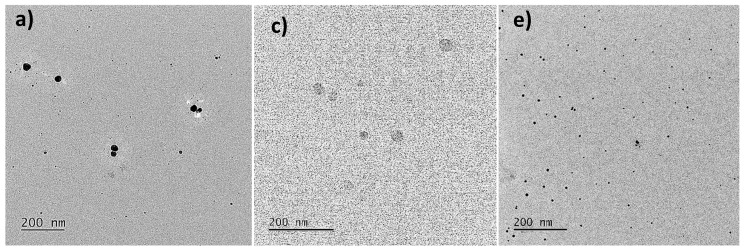
Transmission electron microscopy (TEM) images of AgNPs synthesized in reaction mixtures of different pH. (**a**) and (**b**) pH 12; (**c**) and (**d**) pH 10; (**e**) and (**f**) pH 6.

**Table 1 nanomaterials-12-04184-t001:** Total phenolic and flavonoid contents in methanolic extracts of cell suspension cultures and intact plant leaves from *Randia aculeata* L.

Plant Resource Extract	Total Phenolics (mg GAE/100 g DW)	Total Flavonoids (mg QE/100 g DW)
Cell suspension cultures	162.19 ± 27.9 a	122.07 ± 8.2 a
Leaves	198.07 ± 32.1 a	117.70 ± 15.3 a

GAE = gallic acid equivalents; QE = quercetin equivalents; DW = biomass dry weight. Means with the same letter in the same column are not significantly different according to Tukey’s analysis *p* ≤ 0.05. Values are mean ± standard deviation of three measurements.

**Table 2 nanomaterials-12-04184-t002:** Antibacterial activity of AgNP synthesized with cell suspension culture extracts of *Randia aculeata* L. at different pH values.

Bacteria	MIC (µg/mL)				
	AgNPs				Gentamicin **
	pH 6	pH 10	pH 12	AgNPs *	
*Escherichia coli*	3.9	9.7	14.3	10	≤0.62
*Streptococcus pyogenes*	3.9	9.7	14.3	≥10	5
*Pseudomonas aeruginosa*	1.9	2.4	3.5	5	20
*Staphylococcus aureus*	3.9	9.7	14.3	≥10	≤0.62
*Salmonella typhimurium*	3.9	9.7	14.3	10	≤0.62
*Staphylococcus aureus*-MRSA	3.9	9.7	14.3	≥10	20

MRSA: methicillin-resistant *Staphylococcus aureus*; * Reference commercial AgNPs; ** Reference commercial drug.

**Table 3 nanomaterials-12-04184-t003:** Antiproliferative activity of AgNPs synthesized with cell suspension cultures extracts of *Randia aculeata* L. at different pH values.

Cell Line	IC_50_ [µg/mL]				IC_50_ [ng/mL]
	pH 6	pH 10	pH 12	AgNPs *	Taxol *
Hep3b	1.2 ± 0.30	2.4 ± 0.20	4.3 ± 0.20	4.1 ± 0.20	17.1 ± 0.43
HepG2	1.2 ± 0.06	2.8 ± 0.15	4.8 ± 0.62	4.6 ± 0.40	21.4 ± 0.34
HeLa	2.2 ± 0.13	3.1 ± 0.25	5.2 ± 0.08	4.7 ± 0.40	21.4 ± 1.71
A549	3.0 ± 0.38	5.0 ± 0.23	7.6 ± 0.65	6.0 ± 0.70	68.3 ± 5.98
IHH	3.0 ± 0.32	5.7 ± 0.25	9.0 ± 0.71	8.0 ± 1.0	82.8 ± 6.83

IHH (immortalized human hepatocytes), Hep3B and HepG2 (hepatocellular), HeLa (cervical), A549 (lung). * Commercial references.

## Data Availability

Not applicable.

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
