# Peer review of "Green Synthesis of Silver Nanoparticles Using Randia aculeata L. Cell Culture Extracts, Characterization, and Evaluation of Antibacterial and Antiproliferative Activity"

_nanomaterials, 2022, doi:10.3390/nano12234184_

Round 1

Reviewer 1 Report

Comment for authors

The manuscript “Green synthesis and characterization of silver nanoparticles using Randia aculeata L. cell culture extracts and their potential antibacterial and antiproliferative activity” seem interesting but needs modification before acceptance.

*Abstract. The author has to be concise in a way for better insight into the problem along with objectives and output.

*There are some typos and grammatical errors that need to be addressed properly. Present the consistency. Authors are strongly suggested to seek professional help. The expression of many sentences is too long to understand in a manuscript. Concise them into smaller ones.

*Need to br recheck unit consistency, space and format in the whole manuscript, for example, line 117 (25ºC) and line 143 (25 ± 2 °C).

*In Figures 1A and 2B used space between digits and concentration.

* Why the incubation has an effect on the increment of spectra. Used ref as proof.

*Figure 4 used the same dimension scale bar for all figures A-F for better understanding and comparison.

*Need the pictorial proof as the implication of nanomaterial in the cancer cell. The table does reveal the authenticity of the experiments. Need confocal images or cell count results (graphical images).

*How does the author compare the novelty and comparison statement from the previously published review paper?

*There should be a comparison with the standard method.

*How about the practical application and impact of research?

Author Response

Responses to Reviewer 1

Point 1:

The manuscript “Green synthesis and characterization of silver nanoparticles using Randia aculeata L. cell culture extracts and their potential antibacterial and antiproliferative activity” seem interesting but needs modification before acceptance.

*Abstract. The author has to be concise in a way for better insight into the problem along with objectives and output.

Response 1: 

Abstract was rewritten and reviewed by a native English speaking.

Point 1:

*There are some typos and grammatical errors that need to be addressed properly. Present the consistency. Authors are strongly suggested to seek professional help. The expression of many sentences is too long to understand in a manuscript. Concise them into smaller ones.

Response 2:

The manuscript was reviewed by a native English speaking. Certificate is attached.

Point 3:

*Need to br recheck unit consistency, space and format in the whole manuscript, for example, line 117 (25ºC) and line 143 (25 ± 2 °C).

Response 3:

There was a mistake type. This has been corrected throughout the manuscript. Lines 98, 114 and 151.

Point 4:

*In Figures 1A and 2B used space between digits and concentration.

Response 4:

This was done. Currently Figure 2. Pag. 6.

Point 5:

* Why the incubation has an effect on the increment of spectra. Used ref as proof.

Response 5:

Perhaps there is a confusion in the use of the term "incubation". We use a controlled temperature to avoid bias in our results because of this factor, since in our work place the temperature varies during the day and night. Therefore, we use a temperature-controlled grow room at 25 ± 2°C for better control.

The intensity of the spectrum is not given by the incubation (25 ± 2°C) but is due to the time the reaction mixture was kept under these conditions, where finally no significant changes were observed between 24 and 48 h (Figure 2b). This is explained in the text, and we cite various references. Lines 247-269.

In addition, one of the first characterization techniques carried out on metallic nanoparticles is UV-Vis spectroscopy. Our synthesis proposal involves the use of cell culture extracts containing reducing agents such as polyphenols, among others. These reducing agents act gradually (up to 48 h) until all or most of the silver ions in the precursor are reduced and metallic nanoparticles are obtained. Similarly, the intensity of the spectrum can be used to qualitatively interpret the amount of nanoparticles, that is, the more intense the absorption curve, the greater the amount of nanomaterials. This explanation and references in the text. Lines 247 to 257.

Point 6:

*Figure 4 used the same dimension scale bar for all figures A-F for better understanding and comparison.

Response 6:

This was done. The images in Figure 4 were rearranged and the scale bars were modified for better understanding. Currently Figure 5.

Point 7:

*Need the pictorial proof as the implication of nanomaterial in the cancer cell. The table does reveal the authenticity of the experiments. Need confocal images or cell count results (graphical images).

Response 7:

We appreciate your insightful comments. However, the objective of this study was not to determine the mechanism of action of AgNPs or to know how they are incorporated into cells. Only evaluate the cytotoxic activity. This part is interesting, but those studies could be done in future studies.

In our work, the main objective was to obtain an extract from a Randia aculeata cell culture evaluating pH and time factors, then determine if AgNPs had anticancer effect or cytotoxic activity. In fact, our study is novel because there are no studies on suspension cultures of any plant that has been used for these purposes. However, we could consider your suggestion in other future works.

Additionally, we are adding as Supplementary Material (Figure S2) the microscopic images of the Hep3B cancer cell line, which was the most sensitive to the AgNPs treatment, where morphological changes, growth inhibition, rounding, cell detachment are clearly observed, which according to Hussein et al., 2020 (https://doi.org/10.1007/s10811-019-01905-7) and Husni et al., 2015 (https://doi.org/10.1016/S2221-1691(15)30013-7) are characteristic effects of the cytotoxic effect of a compound. This has been incorporated in the text. Lines 499-501.

Point 8:

*How does the author compare the novelty and comparison statement from the previously published review paper?

Response 8:

An apology, we did not understand to which review article you are referring. But we believe that it refers to other silver nanoparticle synthesis papers. It is true that there are several papers published using physical methods and plant extracts, however, there are few studies where extracts derived from biotechnology cultures have been evaluated as in our study. Our study has the advantage that we can obtain extracts from plant cell cultures throughout the year under controlled conditions, but we do not use extracts directly from plants since the objective of our working group is to use sustainable cultures without damaging the ecosystem, through plant biotechnology. In fact, to our knowledge, it is the first study using a cell suspension culture extracts used as reducing agents for nanoparticles synthesis. We highlight this in the abstract and the main text. Lines 20-22, 32-33, 59-63.

Point 9:

*There should be a comparison with the standard method.

Response 9: As we mentioned before, our study was to obtain AgNPs in a sustainable way or through green synthesis, and even more, using extracts from a cell culture of a plant under study Lines 20-22, 32-33, 59-63. For this reason, we believe it is not appropriate to use a standard method, since, as is known, these methods require chemical agents, high temperatures, vacuum conditions, are expensive and dangerous. This is highlighted in lines 42-45. But our goal was to use a clean method.

Point 10:

*How about the practical application and impact of research?

Response 10:

This has already been answered in points 7, 8 and 9. We use cell suspension culture of Randia aculeata as a sustainable resource using plant biotechnology to produce AgNPs. To our knowledge, this is the first study to be reported in this regard, therefore, our research clearly impacts the progress of green synthesis using plant biotechnology. Additionally, our pharmacological studies clearly show that the AgNPs obtained are effective, then, they can be applied against cancer cells or pathogenic bacteria. In fact, this information is highlighted in the Conclusions section.

Reviewer 2 Report

."Aqueous extracts and AgNO3 solutions were prepared every day ............." The language and synthesis procedure is not clear understanding, why need everyday preparation, how long it was used etc.? 

Under this "2.3. Synthesis of AgNPs" many unnecessary subheadings are included which are really confusing. Those subheadings should be eliminated. 

Sections 2.7.1 and 2.7.2 should be merged and need suitable headings for that. 

under "2.8" no need for any other subheadings.  

"3.2. Optimization of AgNPs Synthesis 224 3.2.1. Effect of AgNO3 Concentration"....and other subheadings are very confusing. 

In text needs more discussion about Figure-1, 2, why is absorbance increasing and decreasing, and their significance, etc? 

Figure-4, is TEM images only. Authors should supply HRTEM images, with lattice fringes or "d" spacing for confirmation of Ag. 

XRD, XPS, and FTIR spectral data are very important those should include to identifying the original materials etc.

Discussions under this " 2.4. Characterization of AgNPs" need to be reorganized with simple language and distinctly. 

Overall the manuscript needs many other characterizations data along with rewritten English language. 

Author Response

Responses to Reviewer 2.

Point 1.

"Aqueous extracts and AgNO3 solutions were prepared every day ............." The language and synthesis procedure is not clear understanding, why need everyday preparation, how long it was used etc.? 

Response 1:

We meant that the AgNO3 solutions were prepared on the same day in which the AgNPs synthesis process was carried out. Similarly, the cell biomass was harvested, and the aqueous extraction process was carried out on the same day to use fresh extracts. Nevertheless, the paragraph was rewritten for clarity. Lines 92-96.

Point 2:

Under this "2.3. Synthesis of AgNPs" many unnecessary subheadings are included which are really confusing. Those subheadings should be eliminated. 

Response 2.

This was done. Unnecessary headers have been removed. And the changes can be seen in Lines 91-122.

Point 3.

Sections 2.7.1 and 2.7.2 should be merged and need suitable headings for that. 

Response 3: This was done. Unnecessary headers have been removed. And the changes can be seen in Lines 158-182.

Point 4:

under "2.8" no need for any other subheadings.  

Response 4: This was done. Unnecessary headers have been removed. And the changes can be seen in Lines 183-313.

Point 5:

"3.2. Optimization of AgNPs Synthesis 3.2.1. Effect of AgNO3 Concentration"....and other subheadings are very confusing. 

Response 5: Unnecessary headings were removed, and some paragraphs were restructured, and some parts are highlighted in yellow. Linea 183-313.

Point 6:

In text needs more discussion about Figure-1, 2, why is absorbance increasing and decreasing, and their significance, etc? 

Response 6: As is well known, one of the first characterization techniques performed on metallic nanoparticles is UV-Vis spectroscopy. Our synthesis proposal involves the use of completely natural reducing agents, from the extracts used, some of which may be polyphenols or reducing sugars, among others. These reducing agents act gradually (up to 48 h) until all, or the vast majority, of the silver ions from the precursor are reduced, and metallic nanoparticles are obtained. In the same way, the intensity of the spectrum can be used to interpret the amount of nanoparticles qualitatively, that is, the more intense the absorption curve, the greater the amount of nanomaterials. This explanation and references can be found in lines 248-255.

Point 7:

Figure-4, is TEM images only. Authors should supply HRTEM images, with lattice fringes or "d" spacing for confirmation of Ag. 

Response 7:

This is correct, in Figure 4 (currently Figure 5) only simple TEM images are shown. Unfortunately, the microscope had some technical problems, for which, we only managed to get images without any on-site analysis like the lattice fringes or d-spacing you rightly mention. However, the scale bar in each image allowed us to measure the size of the AgNPs from each sample by using Paint.net 4.3.12 software. Please consider these images as appropriate.

Regarding the confirmation analysis of the nanoparticle material, an elemental analysis was performed by EDX where the spectrum indicates the presence of silver. The rest of the elements that appear correspond to the grid and the sample holder of the TEM system. We decided to annex the result as Complementary Material (Figure S1). Lines 375-376.

Point 8:

XRD, XPS, and FTIR spectral data are very important those should include to identifying the original materials etc.

Response 8: We agree on the importance of an FTIR analysis, therefore, the characterization of all AgNPs samples by FTIR can be found in section 3.2.2., where the complete analysis of the nanoparticles by this method is described.

Point 9:

Discussions under this " 2.4. Characterization of AgNPs" need to be reorganized with simple language and distinctly. 

Response 9: This was done. Lines 10-122.

Point 10:

Overall the manuscript needs many other characterizations data along with rewritten English language.

Response 10: The manuscript was reviewed by a native English speaking. Certificate is attached.

Reviewer 3 Report

The manuscript entitled, ‘Green synthesis and characterization of silver nanoparticles using Randia aculeata L. cell culture extracts and their potential antibacterial and antiproliferative activity’ reported preparation of silver nanoparticles and their antibacterial applications. I am mentioning some loopholes of this work which should be accounted before publication;

1.      Several articles are already reported on such green synthesis of silver nanoparticles. Then what are the novelties of this? Mention is needed.

2.      The reaction mechanism is not clear. It will be better if the author put one graphical to demonstrate it.

3.      Did the author measure the size variation? What is the particles’ size distribution?

4.      The size variation should be given from TEM as well as DLS.

5.      Is there any peak shift in Fig. 2b?  

6.      How the size of the particles has been affected by the medium pH? Better to include references and discuss.

7.      Some articles have genuine relevance and significance which could be referred for better literature review: https://doi.org/10.1016/j.nanoso.2018.05.002; https://doi.org/10.1016/j.snb.2017.06.068; DOI: 10.1039/C5NJ03409D; https://doi.org/10.1080/19430892.2012.706103.   

Author Response

Responses to Reviewer 3.

Point 1.

The manuscript entitled, ‘Green synthesis and characterization of silver nanoparticles using Randia aculeata L. cell culture extracts and their potential antibacterial and antiproliferative activity’ reported preparation of silver nanoparticles and their antibacterial applications. I am mentioning some loopholes of this work which should be accounted before publication;

1.Several articles are already reported on such green synthesis of silver nanoparticles. Then what are the novelties of this? Mention is needed.

Response 1.

It is true that there are several articles reporting the green synthesis of silver nanoparticles, but most of them use extracts obtained directly from the plant or from organic waste, including fruits or plants edible, which is not entirely sustainable. Our study stands out because we use extracts obtained from a biotechnological culture, i.e., a cell culture of a plant under study in our working group. Moreover, our study has the advantage that we can obtain extracts from plant cell cultures throughout the year under controlled conditions, but we do not use extracts directly from plants since the objective of our working group is to use sustainable cultures without damaging the ecosystem, through plant biotechnology. In fact, to our knowledge, it is the first study using a cell suspension culture extracts used as reducing agents for nanoparticles synthesis. We highlight this in the abstract and the main text. Lines 20-22, 32-33, 59-63.

Point 2:

2.The reaction mechanism is not clear. It will be better if the author put one graphical to demonstrate it.

Response 2:

This was done. A general scheme was added in the text. Figure 1. Pag. 5.

Point 3:

  1. Did the author measure the size variation? What is the particles’ size distribution?

Response 3:

We tried to make measurements to measure the particle size distribution because there were some problems with the equipment, and we couldn't take enough images to make those measurements. However, this distribution was obtained using the DLS technique and can be found in section 3.2.1.

Point 4.

  1. The size variation should be given from TEM as well as DLS.

Response 4

As indicated above, the size distribution was obtained using only the DLS technique (section 3.2.1). The TEM images only gave us information on the shape and some sizes of AgNPs because of pH.  But in future studies we will consider your valuable comments.

Point 5.

  1. Is there any peak shift in Fig. 2b?

Response 5: The resonance plasmon positions correspond to each absorption spectrum of each nanoparticle sample. The main difference between each sample is the concentration of the precursor agent, AgNO3. Therefore, we do not consider making any type of adjustment or shifting of these spectra.

Point 6:

6.How the size of the particles has been affected by the medium pH? Better to include references and discuss.

Response 6: This was added in the text of the manuscript. Lines 331-348.

Point 7:

  1. Some articles have genuine relevance and significance which could be referred for better literature review: https://doi.org/10.1016/j.nanoso.2018.05.002; https://doi.org/10.1016/j.snb.2017.06.068; DOI: 10.1039/C5NJ03409D; https://doi.org/10.1080/19430892.2012.706103.

Response 7:  We have reviewed the literature and have enhanced the discussion with suggested references. Lines: 418-421 and 478-480.

Round 2

Reviewer 1 Report

the manuscript is improved

Author Response

Responses to Reviewer 1

Point 1. the manuscript is improved

Response 1. Thank you! This was done.

Reviewer 2 Report

The review is not satisfactory and lacks very important data like XRD, XPS, HRTEM, etc. 

Author Response

Responses to Reviewer 2.

Point 1. The review is not satisfactory and lacks very important data like XRD, XPS, HRTEM, etc

Point 2. We believe that it would have been desirable to use HRTEM, XRD, XPS to analyze the AgNPs, but our equipment is limited; however, we found several reports where it is acceptable to use only UV-vis, TEM, LSD, FTIR and EDX to characterize the nanoparticles as was done in our study. Furthermore, TEM is widely reported as suitable for characterizing AgNPs and is reported in prestigious journals, for example: https://doi.org/10.3390/nano12224052; https://doi.org/10.3390/nano12193484; https://doi.org/10.3390%2Fijms17091534, therefore, it was not necessary to use an HRTEM in this study. However, in our study, the images obtained by TEM are adequate (Figure 5). On the other hand, our study not only focused on the characterization of AgNPs, but it was a study that starts from the use of plant biotechnology as a sustainable biological resource for the clean synthesis of AgNPs, while the characterization and the biological evaluations were an important complementary part to strengthen our study. Therefore, we consider that our characterization studies are adequate. Then, we do not doubt that in future studies we will be able to make a more specific characterization. 

Reviewer 3 Report

This can be published in its present form

Author Response

Responses to Reviewer 3.

Point 1. This can be published in its present form

Response 1. Thank you!.